# Transformation Path of Ecological Product Value and Efficiency Evaluation: The Case of the Qilihai Wetland in Tianjin

**DOI:** 10.3390/ijerph192114575

**Published:** 2022-11-06

**Authors:** Hang Yu, Chaofeng Shao, Xiaojun Wang, Chunxu Hao

**Affiliations:** 1College of Environmental Science and Engineering, Nankai University, 38 Tongyan Road, Jinnan District, Tianjin 300350, China; 2Chinese Academy of Environmental Planning, 8 Anyang Square, Chaoyang District, Beijing 100012, China

**Keywords:** wetland ecosystem, ecosystem services, ecological products, value transformation path

## Abstract

In order to protect wetland resources, China has developed wetland conservation policies and has made significant conservation investments, but there is still a lack of pathways for the conversion to economic value, making it difficult to meet the demand for continuous investment. We have explored a sustainable ecological conservation mechanism using the Seven Mile Sea as a case study, so that ecological conservation costs can be transformed into economic development behaviors and ecological benefits and socio-economic development can be integrated. This paper assesses the ecological product value of the Qilihai Wetland based on the ecosystem service function value assessment method, which designs the realization path of ecological product value and predicts the value transformation efficiency. The results show the following: (1) The total value of ecological products in the study area is CNY 569.06 million (USD 78.36 million), and the main sources of value are plant products in the supply service and water purification functions in the regulation service, accounting for 54.05% and 26.10% of the total, respectively. (2) The predicted value realization of ecological products, ideally, is CNY 689.65 million (USD 94.96 million), with a value realization rate of 111.60%. Considering the management policy restrictions in different areas of the Qilihai Wetland, the actual value realization volume is CNY 391.94 million (USD 53.97 million), with a value realization rate of 63.42%. (3) Owing to the restriction of the development policy of supply services and cultural services, the value realization path mainly contains two types: one is to drive the development of supply services and cultural services in the surrounding areas, along with product premiums, to realize value transformation. This path is mainly aimed at the supply of local characteristic products and the development of tourism. The second is to realize the value of regulating services through ecological compensation and ecological equity trading. This path is mainly for the adjustment and support services in the core area and buffer area. According to the pre-accounting results, the contribution rates of the two paths are 62.25% and 37.75%, respectively. The second path should be further effectively developed to improve the contribution rate. This study helps assess the ecological value and important ecological elements of the Qilihai Wetland to ensure effective protection and development of important ecological resources and to achieve the sustainable development of wetland resources. It provides a reference for exploring feasible paths to realize the value of ecological goods.

## 1. Introduction

Wetlands are transitional zones between terrestrial and aquatic ecosystems and are known as the “kidneys of the earth”. Wetlands have many wildlife resources and are among the most biodiverse ecological landscapes in nature, and they are also the most valuable ecosystems in the world, with a significantly higher value per unit area than other ecosystems. The second national wetland resources survey shows that China has 53,602,600 hectares of wetlands, accounting for 5.58% of the total land area. Compared to the results of the first survey 10 years ago, the area of wetlands in China had decreased by 3,396,300 hectares by the same metrics, with a reduction rate of 8.82% [1]. Although there has been further improvement in the area of wetland protection, the pressure on the wetlands under threat has continued to increase. Ecological products and their use by humans are major reasons for the significant reduction in wetland area and the increase in the threatening pressure on wetlands [2]. Urbanization and the conversion of natural areas to agricultural land are among the most important threats to the value of wetland ecosystems [3]. The severity of these impacts depends on many factors, such as the vulnerability of the ecosystem, the severity and magnitude of the change, and the sensitivity of the flora and fauna. Wetlands with a wide range of aquatic and plant species are among the most vulnerable ecosystems, especially in response to land-use changes. Spatial and temporal differences in human-induced land-use changes and climate change lead to regional differences in ecosystem services. Scientific assessment of changes in the value of the ecological service functions of wetlands and their influencing factors is of great theoretical value and practical significance for further understanding the evolution of wetland ecosystem service functions, and can provide a basis for implementing ecological protection compensation systems, managing wetland parks, etc.

The Qilihai area has a unique importance and ecological value. Qilihai Wetland is a part of The Tianjin Ancient Coast and Wetland National Nature Reserve. The main protected objects are wetlands and their ecosystems. The total area of the reserve is 344.14 km^2^, including 44.85 km^2^ of the core area, 42.27 km^2^ of buffer area, and 257.02 km^2^ of the experimental area. Qilihai Wetland is a typical area of biodiversity with a wide variety of biological species; it plays an important role in regulating the local climate and purifying the surrounding environment; it also has a unique ecological landscape, rich wildlife resources, and profound cultural heritage, and is an important migration destination and midway point for migratory birds along the sea in East China [4].

It is of great ecological and economic significance to promote the realization of the value of ecological products in the Qilihai region. Turning ecological benefits into economic benefits, promoting the sustainable development of the area around the Qilihai Wetland, and realizing the win–win situation of protecting wetlands and increasing local income have become important measures to explore a new path for the protection and construction of ecological environments.

The concept of “ecological products” was first proposed by Chinese scholars in the 1990s. The specific definition of this concept has not yet been unified, with suggestions including “the natural elements that come from the natural ecosystem to regulate ecological balance and maintain environmental comfort” [5], “the collection of natural elements and business products and services formed by the joint action of natural forces and human labor forces” [6], etc. In recent years, research related to the value realization of ecological products has gradually become a hot topic. In particular, with the gradual research on “ecosystem services”, scholars have gained a deeper understanding of the concept of “ecoproducts” [7]. Ecological products are considered to comprise the collection of ecological services and final material products that humans obtain from nature [8,9,10]. Although the expressions vary, the definition has largely converged, referring to the sum of goods and services that humans obtain from ecosystems. The value realization process of ecological products refers to the release of the value embedded in ecological products, and the mechanism of this realization process is currently the focus of research in China in this field. The realization of value is the process of internalizing its external value characteristics. Therefore, the process of realizing the value of ecological products is the internalization of externalities by the government and the market through a market-based transaction system or nonmarket management [11]. Considering ecological products in the social and economic system, their value realization process requires the realization of value creation and value addition from production to circulation, consumption, and completion of transactions. From the perspective of production and management, ecological products can be incorporated into the four links of production: distribution, exchange, and consumption in the social production process, and ecoindustrialized management is the process of ecological products’ value production and realization [12].

Ecological products refer to the end products or services provided by biological production to human society for use and consumption, including human welfare or benefits such as ensuring the living environment, maintaining ecological security, providing raw materials, and spiritual and cultural services. These are the necessities of life that are parallel to agricultural and industrial products, and meet the needs of humans for a better life [13].

Combined with practice and exploration, several typical modes of realizing the value of ecological products have been preliminarily formed, including ecological protection compensation, ecological industry development, transactions of ecological rights and interests, resource property rights transfers, ecological carrier premiums, resource quota transactions, ecological restoration, etc., (see Table 1 for details).

The typical modes of realizing the ecological product value of wetland ecosystems include ecological sightseeing, popular science education, urban leisure, and compound development (see Table 2 for details).

The concept of “eco-products” is new to China and was first formally introduced in the National Zoning of Main Functions released in 2010. Its specific content and classification are highly relevant to the concept of “ecosystem services”, and “ecological products” are regarded as products with ecological attributes that are produced by a human society based on “ecosystem services”. Accounting for the value of ecosystem services is a means of measuring the value of ecological products, which is important in cognition and decision-making [14], and helps to achieve sustainable development [15]. Domestic and international scholars have discussed the various benefits that people receive from ecosystems in terms of monetary and service markets, and are constantly exploring and researching new valuation methods [16]. In the 1990s, Contanza et al. classified the value of global ecosystem services and grouped ecosystem service functions into 17 categories [17] on the basis of which ecosystem service functions were assessed. Wallace classified ecosystem services into four categories based on human needs [18]. To promote the conservation and sustainable use of ecosystems, the United Nations released the Millennium Ecosystem Assessment (MA) report and the Global Biodiversity and Ecosystem Services Assessment report. These reports underscore the importance of nature’s benefits to sustainability; however, 60% of the ecosystem services covered in the Millennium Ecosystem Assessment are being degraded or used in an unsustainable way by humans [19]. Many ecosystem services have been degraded to increase the supply of other services, such as food, and the tradeoffs are often passed on to other groups of people or deferred to future generations. This degradation of ecosystem services has become a major obstacle to the United Nations’ Millennium Development Goals (MDGs), so increasingly more scholars in China and abroad are focusing on an integrated, multiscale assessment of various ecosystems around the world, and the importance of ecosystem service functions is gradually gaining attention. The value of ecosystem services is not only a reflection of changes in the physical stock of ecosystems, but also an estimate of the value of ecosystem products consumed by humans in specific socioeconomic contexts, and this assessment is based on specific economic and social development levels. The United Nations continues to guide global sustainable development with the more comprehensive and specific 2030 Sustainable Development Goals (SDGs), based on the consolidation of the existing achievements of the Millennium Development Goals (MDGs), which are of great significance in the realization of pathways to rebuild degraded ecosystems and carry out sustainable development of the region. Domestic research on the assessment of ecosystem service functions started late and mainly focused on static and dynamic studies of ecosystems to assess the value of ecological service functions.

The goal of this paper is to study the connotation of ecological products and the path of transforming ecological products into ecological values, and to discuss how to effectively integrate the relationship between conservation and economic development. Thus, the ecological benefits formed by ecological protection and the socio-economic benefits can be connected to form a sustainable ecological construction mechanism and model with active participation from everyone. In this process, we also combined the relatively mature ecological service function value accounting method with the resource and environmental endowment conditions of Tianjin, made local optimization and improvement, and adjusted the corresponding key parameters.

## 2. Methodology

### 2.1. Technical Process

In order to explore an effective way to realize the value of ecological products, the following technical processes are assessed in this paper (Figure 1). It mainly includes three parts: 

The first process is the determination of the study area and the investigation of its basic condition. The research area should be determined according to the demand for realizing the value of ecological products. After the research area is determined, the basic ecological conditions of the area shall be assessed through field surveys and reference materials. Using this approach, the project list of important ecological products is determined. The next step is to calculate the value of ecological products in the study area. According to the list of ecological products and the actual condition of the region, the selection of accounting methods and the determination of parameters are completed. The value of ecological products is calculated based on the determined methods and parameters.

The paths to realize the value of ecological products are diverse, including both government-led and market-led ways. In fact, not all paths are applicable to the selected area. Therefore, it is necessary to choose the path according to the type of ecological products and the actual development situation of the region. Then, according to the selected path, the value realization effect of ecological products is evaluated. After evaluating the effect of the value realization path, a comparative analysis is needed to further optimize the path.

### 2.2. Selection of Study Area

The development and protection of wetlands in China is of great significance. At present, significant protection work has been carried out for wetlands [20]. It is necessary to realize the value transformation of protection costs and explore sustainable value realization mechanisms. Therefore, it is urgent to explore the paths to realize the value of ecological products in order to meet the demand for sustainable investment. For the purpose of providing a reference for the above needs, this study takes Qilihai Wetland as a case example.

In 1992, the State Council approved the establishment of the Tianjin Ancient Coast and Wetland National Nature Reserve (Figure 2). This reserve is rich in natural resources, with a complete wetland system and high biodiversity. The abundant plant resources play important roles such as maintaining water sources, improving water quality, providing flood storage and drought prevention, regulating climate, and maintaining biodiversity [21]. The habitat function value of Qilihai Wetland is very important; it is home to more than 180 kinds of rare and protected birds, which are of great significance to the study of land and sea changes and coastal wetland ecosystems. In addition, it is also a characteristic tourism and education resource. The Qilihai National Wetland Park—a national nature reserve in Qilihai Wetland that was constructed by the government with an investment of CNY 1 billion—was officially opened in 2013. In 2014, the reserve was included in the city’s ecological land protection red line by the Tianjin Municipal Government. According to the Administrative Provisions of Tianjin on Permanently Protected Ecological Areas, tourism activities are not allowed within the red lines of the permanently protected ecological areas, so Qilihai Wetland Park has been closed since September 2015 [22].

Since the areas with sensitive ecological environments and facing serious impacts of human activities are generally the core area and the buffer zone, and the current protection and utilization measures and regulations for the Qilihai Wetland are mostly concentrated there, this study focuses on the core area and buffer zone, and expands outward to a certain extent [23]. The scope of the ecological impact assessment reflects ecological integrity, comprehensively considering climate, hydrology, ecological units, geographical units, and other factors, and the extension range is ~1–3 km. In order to meet the requirements of scenario design for realizing the value of ecological products, it is also necessary to properly include the surrounding developed and built land.

Based on these principles, the buffer zone was extended for ~2 km to form the final research area. The final research area included all of the core areas, buffer areas, some of the experimental areas, and a small number of areas outside the protected areas. Among them, all human activities are prohibited in the core area. All tourism, production, and business activities are prohibited in the buffer zone, although tourism and other development activities are permitted in a pilot area. There are no regulations outside the reserve. The selected study area is shown in Figure 3.

### 2.3. Design of the Path to Realize the Value of Ecological Products

There are two kinds of logic to realize the value of ecological products—one is the logic of “transformation” and the other is the logic of “protection”. Accordingly, there are two different ways to realize the value of ecological products: one is marketization, the other is government regulation. The logic of “transformation” needs to make more use of the market mechanism to make the hidden value of ecological products appear in the market. The logic of “protection” requires the government to play its regulatory role, implement ecological compensation and other means, so that the external value of natural protection can be converted into the real value of monetization. The choice of value realization path needs to be combined with the actual situation of the research area In the process of value realization path design, we took the typical model of ecological product value realization, relevant cases, and policies issued by relevant national ministries and commissions as references. We then optimized and designed the realization scenario of the ecological product value in the study area, including three modes—ecological industry, ecological compensation, and ecological transaction.

#### 2.3.1. Ecological Industry

The development of the ecological industry is the mode of the exchange value of operational ecological products through market mechanisms; it is the process of industrialization of ecological resources, and is the value realization mode with the highest degree of marketization at present. In the study area, ecological management of agriculture and forestry drives the development of ecotourism, so as to realize the value appreciation of agriculture, forestry, and tourism products, drive the increase in land prices in the region, and demonstrate the value of ecological products. Based on the actual condition of the Seven Mile Sea wetland, the following four pathways are planned.

(1) Ecological agriculture:

The original special products in Qilihai Wetland include reeds, whitebait, and purple crab, which are the special ecological agricultural products in the region. *Phragmites australis* is the dominant plant in the Qilihai Wetland, and its coverage in the east and west of the area reaches 60–80%. Reeds can be used for the manufacture of paper and textiles, with high raw material production value. At the same time, during the growth of reeds, their roots can absorb nitrogen and phosphorus from water, and perform many functions, such as climate regulation, water purification, and maintaining biodiversity. The production of silver fish, purple crab, and other foods in the Qilihai Wetland is huge and can bring great economic value. 

(2) Ecological premiums:

A typical mode of direct carrier premiums of ecological products is to improve the regional ecological environment and increase the supply capacity of ecological products, driving regional land and house property appreciation. Through the overall planning of the development and utilization mode of the study area, the ecological advantage of the beautiful environment can be transformed into the direct economic advantage of land and house property appreciation.

(3) Ecological forestry:

The study area contains a large range of cultivated land. The cultivated land is planned to be transformed into economic forests, which could provide some economic benefits in terms of physical products and beautify the environment. Tianjin has a clear pattern of economic forest development. The eastern coastal area is a wine-grape-producing area; northwest Tianjin is a protected fruit-cultivation area, and there is a characteristic high-quality fruit production area in the northern mountain area of Jixian County. In this study, only the cultivated land area around the study area was converted into an economic forest, and the economic benefit analysis was carried out according to the average levels of Tianjin. No specific details—such as forest type and scale—were planned. 

(4) Ecotourism:

The development path of ecotourism is based on the natural landscape resources of the Qilihai area. Ecotourism is developed around rural characteristic landscapes to achieve wetland protection and farmers’ income increase. Optional development projects include wetland culture education and training, live-action performances, and main Through tourism revenue to achieve the conversion of wetland ecological protection costs to economic value.

#### 2.3.2. Ecological Compensation

Ecological compensation can use multiple approaches, including government purchases, transfer payments, ecological compensation, and other financial transfers. It can also be based on cross-regional horizontal transfer payments by watershed. Based on the actual situation of the Qilihai region, it is suitable to use direct government funding for compensation. Direct government funding for ecological compensation is used as a specific path for this model.

#### 2.3.3. Ecological Transactions

There are two main ways to implement the ecotrading model. One is to realize the capitalization of natural resources through the concentration of fragmented property rights, and then realize profits through rational management. The second is the market-based trading of important ecological resources such as wetland areas and carbon sinks in the region under the premise of a sound trading mechanism. Based on the second realization approach, this study identifies two trading paths: resource quota trading with wetlands as the carrier, and ecological equity trading with carbon sinks as the carrier.

### 2.4. Ecological Product Value and Economic Benefit Accounting

#### 2.4.1. Value Accounting

The value evaluation methods of ecological products include material quality evaluation, energy value analysis, and value quantity evaluation. This research mainly adopts the value evaluation method. Such methods express the value of ecosystem services in the form of currency, which is intuitive and conducive to attracting the attention of the government and the public. They are divided into direct market methods, simulated market methods, and alternative market methods [24,25,26,27,28], including the market value method, protection cost method, shadow engineering method, conditional value method, etc. See Table 3 for the accounting method.

(1) Supply services:

The value of the supply services is calculated by the market value method. The supply of material products in the study area is mainly reflected in aquatic products and plant products. Aquatic products mainly include whitebait and purple crab, which are characteristic species of the Qilihai Wetland; according to relevant statistics, their annual output reaches 1.2 × 10^6^ kg and 5 × 10^5^ kg, respectively. The prices of silver carp and purple crab in this study refer to the market price. Plant products mainly include reeds and grain. The direct market method is adopted for the evaluation of reeds’ value. Calculated according to the unit yield of 6500 kg/hm^2^ and the market price of 300 CNY/t, the harvesting area is 50% of the total production area. The grain output refers to the average output per unit area of Tianjin, and the price refers to the market price.

(2) Atmospheric regulation: 

According to the photosynthesis equation of plants—6CO_2_ + 12H_2_O = 6 (CH_2_O) + 6O_2_ + 6H_2_O—plants need to absorb 1.63 g of CO_2_ and release 1.19 g of O_2_ for every 1 g of dry matter produced. The carbon dioxide absorption and oxygen release values of wetlands can be calculated according to this equation. The reed harvest in the wetland of the study area is the main source of dry matter production, and the annual yield of reeds was taken for the calculation. The price of carbon fixation and oxygen release was determined in combination with domestic afforestation costs and industrial oxygen production standards. The carbon fixation value is 625.45 CNY/t, which is the arithmetic mean value of China’s afforestation cost of 260.90 CNY/tC (constant price in 1990) and the internationally accepted Swedish carbon tax rate of 150 USD/tC (the exchange rate between CNY and USD was calculated at CNY 6.60 at the end of 2017). The price of industrial oxygen production is 0.4 CNY/kg.

(3) Flood regulation and storage:

According to the shadow engineering method, the value of flood regulation and storage function can be estimated by using the cost of the storage capacity and the maximum flood volume that can be stored in the Qilihai Wetland. According to the national reservoir construction investment from 1997 to 2009, considering the price rise index and the actual situation of the Qilihai region, the average construction price of 6.1 CNY/m^3^ per unit reservoir capacity was determined. In combination with the current situation and development planning of Qilihai Wetland, the variation in the water level was calculated as 0.3 m. The water storage was determined by combining the wetland area with the remote sensing map.

(4) Water purification:

Wetlands have powerful water purification functions and provide a natural space for humans to manage pollution. Wetland ecosystems use physical, chemical, and biological synergies to degrade and purify pollutants in water through filtration and adsorption by soil, absorption by plants, and degradation by microorganisms. Pollutants can also be completely removed from the system by regular harvesting of aquatic plants [29]. Reeds in the waters of the study area are regularly harvested aquatic plants, so the value of water purification can be estimated from reed production. According to Costanza [19], the value of water purification per unit area of wetland ecosystem is 4177 USD·hm^−2^·a^−1^, which is equivalent to 26,394 CNY·hm^−2^·a^−1^. The value of water purification services was calculated by combining the wetland area in the study area.

(5) Production and maintenance of biodiversity:

The Qilihai Wetland is one of the richest and most distinctive areas in terms of biodiversity. Its large areas of reed swamps, shallows, and ponds provide natural derivation sites for wildlife. The special habitat of the wetland provides a rich source of food and a good place to nest and avoid predators for various wading birds. There are 153 species of wild plants in 41 families in the wetland, accounting for 93% of the wetland plants in the coastal area of Tianjin. Among them are a large number of ornamental plants. In addition, there are 108 species of traveling birds in the wetland. The numerous wildlife resources in the Qilihai Wetland play an important role in maintaining the ecological balance of nature. Its biodiversity value can be obtained from the ecological service value of biodiversity per unit area and the total wetland area. The biodiversity value was assessed using the annual ecological benefit of 439 USD/hm^2^ calculated by Costanza et al. from the results of the bioshelter service value per unit area of coastal zone salt marsh wetlands [22], before integrating the biodiversity maintenance service value per unit area of wetlands in China as 2212.2 CNY·hm^−2^·a^−1^ [30]. The arithmetic mean of the above two calculations was taken as the average biodiversity maintenance service value per unit area of the wetland, which was 2554.8 CNY·hm^−2^·a^−1^.

(6) Maintain soil nutrients and reduce waste soil:

The accounting of this value needs to be based on the acquisition of soil conservation data. The RUSLE model was used to simulate field measurements to obtain the corresponding data [31]. We evaluated the intensity of the ecosystem’s soil conservation function using rainfall, slope length, vegetation, and soil. Then, we obtained values of the soil conservation function and waste reduction function of wetlands using the cost substitution method. The estimation of soil conservation was based on the modified universal soil erosion equation (RULSE) calculation:(1)Q=R×K×L×S×(1−C×P) 
where *Q* is the amount of soil conservation (t·hm^−2^ a^−1^), *R* is the rainfall erosion force factor (MJ·mm·hm^−2^·h^−1^·a^−1^), *K* is the soil erodibility factor (t·h·MJ^−1^mm^−1^), *L* is the slope length factor (Dimensionless), *S* is the slope factor (Dimensionless), *C* is the vegetation cover factor (Dimensionless), and *P* is the soil and water conservation factor (Dimensionless).

Accounting for the value of soil nutrient retention and waste soil reduction based on obtaining the amount of soil retention. 

The value of maintaining soil nutrients can be estimated by the value of nitrogen, phosphorus, and potassium in the soil. Only ~40% of the nitrogen, phosphorus, and potassium applied as agricultural fertilizers is used by crops each year, with the remaining 60% lost in various forms. The agricultural fertilizers in the area around the Qilihai Wetland are mainly nitrogen and phosphorus fertilizers, while the amount of potassium fertilizer is negligible, so the potassium fertilizer was ignored in our calculations. The soil nitrogen and phosphorus contents of the Tianjin Qilihai Wetland were 1.6 [32] and 0.5 g/kg [33], respectively. Fertilizer prices were based on the market prices of urea and diammonium phosphate to project the prices of pure nitrogen and phosphorus in fertilizers. The annual average agricultural income in the accounting of the value of waste soil reduction was obtained from the Tianjin Statistical Yearbook and the Ninghe District National Economic and Social Development Statistical Bulletin. The soil capacitance in Tianjin is 1.01–1.65 g/cm^3^, and the soil thickness was taken as 0.5 m [34].

#### 2.4.2. Economic Benefit Accounting

##### Ecological Industry

Ecoindustry development is a mode of realizing the exchange value of operational ecological products through market mechanisms and is the process of industrializing ecological resources, which is currently the most market-oriented mode of value realization. In the study area, the ecological operation of agriculture and forestry leads to the development of ecotourism, realizing the value appreciation of agricultural, forestry, and tourism products, while also driving up the prices in the region and realizing the value manifestation of ecological products.

(1) Ecological agriculture:

A more substantial product premium can be achieved through ecological agriculture, organic farming, etc. The premium is mainly focused on supply service products. The reed-farming area was calculated based on the existing reed-planting area in the land-use types in 2021, and the harvestable area was calculated as 50% of the planting area. Owing to the lack of specific farming areas and other factors, the production of silver fish and purple crab was based on the production in 2021.

Influenced by uncontrollable factors such as product characteristics, market rules, and marketing methods, the product premium range is difficult to predict precisely. Referring to the existing practice cases, the premium range of ecological products is 19–3000%, which shows a large uncertainty in the change range. This is related to a variety of factors—such as local characteristics, market laws, and the quality of the product itself—and is difficult to quantify precisely. The average premium rate of “Lishui Mountain Farming” products is 30%. After the development of organic cultivation, the price of Shihuoji tea in Baoxi County, Anhui Province, gradually increased from 200 to 6000 CNY/kg in 2019. At the research level, willingness to pay varies across types of ecofriendly products [35,36,37,38,39]. In this study, a relatively conservative estimate was taken to calculate the value of plant products and aquatic products at a premium rate of 30%. The premium estimation was based on the value of supply services.

(2) Ecological premium:

The city of Zhangzhou has achieved a land premium of CNY 2000 per square meter by attracting enterprises to improve the environment through a public–private partnership approach. Wuyuan Bay in Xiamen, Fujian Province, has increased the supply of ecological products through ecological restoration and comprehensive development, driving up the value of surrounding land premiums. Jiawang District in the city of Xuzhou, Jiangsu Province, has carried out ecological restoration and comprehensive renovation of a coal mining subsidence area. The increase in the supply of high-quality ecological products has driven regional land appreciation, and the residential land price has increased significantly. Owing to the difficulty in specifying the magnitude of land price premiums, estimates were made on the basis of existing cases. The land price increase of more than 300% in Jiawang District (Xuzhou, Jiangsu Province) is based on the results of significant changes in the ecological environment before and after ecological restoration, and is not applicable to the estimation of land premiums for general ecological planning and development. Referring to the commercial property sales in the city of Zhangzhou as of 2018, the premium range of Zhangzhou—i.e., a 15% land premium—was used, which is more appropriate for the Qilihai area. The number of commercial houses for sale in each township in the study area lacks relevant statistical data, and was projected based on the relevant data of Ninghe District. There is an obvious phenomenon that the sale of commercial houses varies with the size of the city, so the number of commercial houses for sale in the study area is estimated by the population size. According to the relevant information, the sales area of commercial properties in Ninghe District in 2020 was 457,800 square meters. The formula for calculating the real estate ecological premium follows:(2)$$V_{h}=\frac{S_{study\_area}}{S_{ninghe}}*S_{h}*a*b$$
where $V_{h}$ is the property premium gain (CNY), $S_{study\_area}$ is the population in the study area (10,000 people), $S_{ninghe}$ is the total population of Ninghe District (10,000 people), $S_{h}$ is the total area of commercial houses for sale in Ninghe District (m^2^), a is the average house price in Ninghe District (CNY/m^2^), and *b* is the premium margin, which is taken as 15% in this study.

(3) Ecological forestry:

The arable land in the study area was integrated into economic forests. The average income level of economic forestry in Tianjin, which adopts both dry and fresh fruit operation methods, was accounted for. The production area of economic forests in Tianjin is 9.3 × 10^4^ hm^2^, with an annual output of 24.1 × 10^4^ t of various kinds of dried and fresh fruits, and an output value of CNY ~900 million. Taking the dry and fresh fruit operations as the planned development path, it can be determined that the income per unit area of economic forest in Tianjin reaches 9677.42 CNY/hm^2^ [40].

(4) Ecotourism:

Typical ecotourism cases were selected to estimate the benefits of developing ecotourism in the Qilihai area based on the average levels of benefits of existing cases. The selected typical cases are shown in the Appendix A. The calculation formula follows:(3)$$V_{t}=V_{at}\times S$$
where $V_{t}$ is the tourism revenue (million CNY); $V_{at}$ is the average of the annual revenue per unit area of the selected typical cases, calculated as 1.98 million CNY/km^2^; and *S* is the area of the study area (km^2^).

##### Ecological Compensation

An ecological compensation model refers to the act of paying the cost of ecological protection labor or limiting the opportunity cost of development by the government or related organizations to the property owners of the regions or ecological resources producing ecological products in the public interest, which is the most basic means of realizing the economic value of public ecological products. In this study, the economic value of the ecological compensation model was determined through direct financial compensation from the government.

The prerequisite for implementing ecological compensation is the need to clarify the value of regional ecosystem services [41]. In the process of calculating regional ecological compensation standards, in order to mitigate the urgent development needs of areas with lower GDP per unit area and reduce the risk of rapid reduction in regional ecological values, the compensation fund input of areas with more urgent ecological compensation needs should be given priority under the situation of limited repayment capacity. Based on this, the ecological compensation standard is reflected by a regional ecological compensation demand intensity coefficient and an ecological value conversion coefficient. The specific calculation formula follows:(4)R=V×k×t
(5)$$t_{i}=2\times \tan^{-1}ESPC/\pi$$
(6)$$ESPC=N_{ESV}/GDP$$
where *R* is the regional total amount of ecological compensation (million CNY); *V* is the regional nonmarket ecosystem service value (million CNY); *k* is the nonmarket ecosystem service value conversion factor, whose value was chosen as 15% with reference to previous research [42]; *t* is the ecological compensation demand intensity (dimensionless); *ESPC* is the ecological compensation priority index (dimensionless); π is the circumference; $N_{ESV}$ is the nonmarket ecosystem service value (million CNY); and *GDP* is the gross regional product (million CNY).

##### Ecological Transaction

The ecotrading model refers to a model in which a region relies on its own resource endowment and realizes the exchange value of quasi-public ecological products—such as resource quotas and ecological rights and interests—through market mechanisms, under the provisions of national policies such as the “balance of occupation”. The scarcity of resources generated by government regulation is a prerequisite. This study identifies two trading paths: resource quota trading with wetland areas as the carrier, and ecological equity trading with carbon sinks as the carrier.

(1) Resource quota transactions:

Market-based trading of wetland areas as an indicator of occupation and replenishment is still in the exploration stage in China, and there are few practical cases. In this study, the value of nonmarket ecosystem services per unit area in the study area was used to represent the trading value of wetlands per unit area, and three land-use types—namely, water, reed, and marsh—were selected for matching trading of wetland resources. The calculation formula follows:(7)$$V_{r}=\frac{N_{esv}}{S}*S_{t}$$
where $V_{r}$ is the resource quota trading value (million CNY), $N_{esv}$ is the nonmarket ecosystem service value (million CNY), S is the area of the study area (km^2^), and $S_{t}$ is the area of wetlands available for trade (km^2^).

(2) Ecological equity transaction

The price of wetland carbon sinks is 58.83 CNY/t, and the amount of carbon sequestered in the study area was determined based on the amount of carbon sequestration function accounted for by the value of the above ecosystem services.

## 3. Results and Discussion

### 3.1. Accounting Results

The results of the ecological product value accounting are shown in Table 4. The value of ecological products in the study area in 2021 was CNY 569.06 million (USD 78.36 million), of which the highest value was that of supply services at CNY 366.77 million (USD 50.50 million), accounting for 64.45% of the total value of ecological products. The lowest value was that of support services at CNY 21.21 million (USD 2.92 million), accounting for 3.73% of the total value of ecological products. The value of regulation services was CNY 181.09 million (USD 24.94 million), accounting for 31.82%. Among the provisioning services, plant products contributed the most to the value of ecological products, at 54.05% of the total value. The largest contribution of water quality purification in regulating services was 26.10%. The overall ranking is Plant Products > Water Purification > aquatic Products > Flood water Storage > Generation and maintenance of biodiversity > Atmospheric conditioning > Soil Conservation > Reduction in Waste Soil. 

The pre-accounting results of the value realization of ecological products are shown in Table 5. The pre-accounting result of the value realization of ecological products was CNY 689.65 million (USD 94.96 million), of which the largest contribution was made by ecotourism, with a value of CNY 330.84 million (USD 45.56 million), accounting for 52.44% of the total value realization. Ecoequity trading had the lowest value realization at CNY 0.31 million (USD 0.04 million), or 0.04% of the total value realization. The overall ranking is ecotourism > ecoforestry > ecological agriculture > ecopremium > resource quota trading > ecological compensation > ecological equity trading.

The results are shown in Table 5.

### 3.2. Value Realization Characteristics

The value realization characteristics of ecoproducts are shown in Figure 4. Because of the current management regulations in the study area, all tourism development projects were cancelled, so the tourism revenue was 0. In order to reflect the changes in the value of cultural services, tourism revenue in 2014 (considering inflation) was used as a proxy term for comparative analysis. The results show that the total value realization rate is 111.60%, realizing the value transformation and value addition of ecological products. Material products have more mature trading markets and trading mechanisms, and 100% of their economic value can be realized through market transactions. The value conversion rate of the supply services in the study area is 49.09%. The lower conversion rate is caused by the changes in product type. The actual types of ecological products supplied in 2021 included reeds, food, and aquatic products. The large arable land area is responsible for the majority of the value of food products. The pre-accounting scenario planning converts arable land into economic forests in order to improve the regional ecological environment and enhance the ecological capital stock, leading to a decrease in the economic value of supply services. However, the increased ecological value is transformed into economic value through the increase in the value of cultural services. The value conversion rate of regulating and supporting services is 73.14%. As an important part of ecological product value, this part of the value is mainly realized in the form of land and house property premiums, compensation, and transactions. Owing to the lack of a mature trading market and trading mechanism, such services rely on special value realization paths, resulting in a low-value realization rate. Under the existing pathway, only the sequestered carbon value has an independent and clear pathway to value realization, i.e., wetland carbon sink trading. Other regulations and support services (e.g., flood storage, water purification, soil conservation, etc.) rely on the resource quota trading model, ecological compensation model, and ecological premium model using the wetland area as a carrier. There are no targeted value realization pathways for each ecological value. Existing pathways do not fully realize the value of regulation and support services. Cultural services have the highest amount of value realization, accounting for 52.44% of the total, and the highest value realization rate, far exceeding the value realization rates of supply, regulation, and support services. On the one hand, this is due to the low intensity of the original tourism development caused by the stewardship policy of the area, which has now banned all sightseeing and leisure projects, and the value of cultural services is not reflected. On the other hand, it also shows the superior ecological condition of the study area, which has high development value and potential for tourism development. The lower value of supply services due to the improvement of the ecological environment, along with the lower efficiency of the conversion of regulation and support services due to the lack of value realization paths, can be better compensated and enhanced by the conversion of cultural service values.

### 3.3. Discussion

#### 3.3.1. Analysis of the Main Ecological Elements

The two largest contributions to the value of ecological products in the study area were the supply of plant products and the value of water purification. Plant product supply is the most important source of the ecological product value in the region, with food supply from cultivated land dominating. This is mainly due to the presence of large areas of cultivated land in the study area. The water purification value of the wetlands comes from the absorption enrichment of aquatic plants. The study area is the largest source of reeds in Tianjin, with an annual production of over 30 million kg. The regular planting and harvesting of reeds provide considerable water purification capacity. The value of aquatic products is also more objective. The silver fish and purple crab in the wetland are the special aquatic products of the Qilihai area and have a high market value.

Ecotourism generated significantly higher economic returns than other paths. Since the calculation of ecotourism benefits in this study refers to the average level of relevant typical cases in China, the level of tourism benefits is a reflection of the domestic average. The significant differences between other paths and ecotourism indicate that under the current status quo, the development of ecotourism has a high rate of return and is a development model with a high efficiency of value transformation. The accounting of the economic benefits of the agroecological model is a premium estimate based on the ecological products already available in the region. The main areas of value realization are the aquatic and reed products produced in the core area, which still show a high contribution rate under a more conservative premium rate estimate (i.e., 30%). 

Referring to other successful cases in China, the premium rate of ecological agriculture and fishery products developed by the ecoindustry has a wide range of fluctuation depending on the market situation, product characteristics, market promotion, and other factors. For example, the pricing of organic fish in Qiandao Lake is more than double the market price. Therefore, the ecological agriculture development model has high development potential and is an important way to realize the value of regional ecological products for future research. 

The ecological premium model in this study refers to the typical direct carrier premium in the form of increasing the supply capacity of ecological products by improving the regional ecological environment, which leads to the appreciation of surrounding land and house properties. The main value realization areas are the land and house properties in the building sites. According to the “Qilihai Ecological Protection and Restoration Plan”, in order to effectively protect the wetland resources, ecological migration may be implemented for some villages and towns in the future, directly affecting the land and house properties carrier premium in the area; there is also an uncertainty in the fluctuation of the housing price market, so the ecological carrier premium of land and house properties should not be used as the main means to realize the value of ecological products in the area. 

The ecoforestry development model uses the average level of economic forestry returns from dry and fresh fruit production in Tianjin as the assessment standard, and changes it on the basis of the original arable land, without considering the influence of ecological premiums for the time being. Its value realization area is the original arable land area. It is also highly implementable and directly connects with the market system, allowing the establishment of branded ecolabeled products. In the future, it has high development potential in combination with the production of dried and fresh fruits and the related forest economy industry in the region.

#### 3.3.2. The Actual Value Realization Rate of Ecological Products Is Lower than the Ideal Situation

The management policies in the study area resulted in a gap between the actual rate of ecological product value realization and the ideal situation. The Tianjin Ancient Coast and Wetlands National Nature Reserve is divided into core areas, buffer areas, and pilot areas. The study area contains all of the core areas and buffer areas, and some of the pilot areas. According to the relevant management regulations, human activities are prohibited in the core and buffer zones, and these two areas occupy ~48% of the study area. Therefore, in practice, the realization of supply services and the value of cultural services are affected to a great extent. All of the aquatic products, reed products, and some arable land in the study area cannot be effectively supplied and traded for products. The value forecast of cultural services is based on the value per unit area, and the amount of value realization needs to be reduced by the same proportion. The actual situation is shown below, with value realization rates of 19.19%, 73.14%, and 354.92%, for supply, regulation and support, and cultural services, respectively. The overall value realization rate is 63.42%. As shown in Figure 5.

In view of the special management policies in the Qilihai region, there are difficulties in realizing the value of supply services and cultural services in the central region, recognizing the value of regulation and support services is the main path to realizing the value of ecological products in the region. However, the pre-accounting scenario is designed in such a way that the value transformation by means of trading and compensation is less efficient. This may be because of the single and inefficient means of trading. The resource quota trading and ecological equity trading markets based on wetland areas and wetland carbon sinks are not yet fully mature, and there may be problems in pricing and other trading mechanisms. In addition, for different kinds of ecological products, there is a lack of targeted value realization paths. Only the value of carbon sequestration has been transformed through the trading of wetland carbon sinks. The functions of flood storage, water purification, and biodiversity generation and maintenance lack independent value realization paths, and there may be inefficiencies in realizing value only through resource quota trading based on wetland areas and the premium effect of land and house properties.

Many nature reserves in the world have a reality similar to that of the Qilihai Wetland. Management policies that severely restrict development practices are designed to guarantee that ecological assets are not diminished. Such policies have a double impact on the ecological products of nature reserves. On the one hand, they enhance the ecosystem service function, i.e., the actual stock of ecological products is richer. On the other hand, they limit the transformation of the ecological value of ecological products into economic value, which is mainly reflected in ecological products highly related to human activities, such as supply services and cultural services. Under such conditions, in order to fully realize the value of ecological products in nature reserves and achieve efficient transformation of ecological value to economic value, the main path is to realize the value of ecological products that do not depend on human development activities—such as regulation and support services—through measures such as compensation and trading.

#### 3.3.3. The Areas for Realizing the Value of Ecological Products Need to Be Further Defined

This study limits the spatial scope of premiums for material goods, land and house properties, and tourism development to the study area. In fact, the scope of premiums and ecotourism development still has the potential to be further extended. Therefore, the full premium of the ecological function of the Qilihai Wetland is not fully reflected in the results of this study. This is one of the reasons for the low-value realization rate. To more accurately account for the full ecological premium of an area, the premium area should first be clearly delineated. Different value realization paths have different premium ranges. How to determine the actual premium range of different value realization paths is a problem that needs to be further studied.

#### 3.3.4. Suggestions for the Development of the Qilihai Wetland

For the special situation of the Qilihai area, the following recommendations are made:

The value realized by reducing the control measures in the core area and buffer zone can be indirectly reflected by enhancing the development opportunities of the surrounding areas. When planning and studying the sustainable development path of the region, we can rely on the good ecological environment of the region and develop the surrounding material supply and cultural service industries to seek the indirect embodiment of the value of the core area and buffer zone. We can then clarify the industrial chains corresponding to various ecological products, increase the value realization path of supply services and cultural services, and increase the value realization rate of ecological products from the industrial perspective. For example, by relying on the excellent supply service and cultural service background ecological assets of the core area and buffer zone, we can develop tourism, education, recreation, and other industries around the region.

The amount of realized value of ecological rights and interests traded in the study area is at a relatively low level. It is recommended to further identify and integrate ecological assets, and to clarify the corresponding tradable ecological products and their functional and value volumes, exploring the pricing model of regulation and support service products, and market trading mechanisms.

We should also attract social groups to participate in ecological compensation, expand the sources of funds, increase the amount of compensation, and explore diversified compensation methods such as ecological management and care of public welfare posts. At the same time, we could explore special ecological compensation for ecological products that cannot be traded in the market at this stage, such as flood storage, biodiversity maintenance, soil conservation, and other services.

## 4. Conclusions

The goal of this paper is to study the connotation of ecological products and the path of transforming ecological products into ecological values, and to discuss how to effectively integrate the relationship between conservation and development. Thus, the ecological benefits formed by ecological protection and the socio-economic benefits can be connected to form a sustainable ecological construction mechanism and model with the active participation of everyone. The results of the study indicate that the value of ecological products in the study area in 2021 was CNY 569.06 million (USD 78.36 million). Under an ideal planning scenario, the ecological product value of CNY 689.65 million (USD 94.96 million) can be realized, with the ecological product value realization rate reaching 111.60%. However, considering the management policy of the Qilihai area, the actual value realization rate will be reduced to 63.42%.

At present, the Qilihai Wetland has good ecological resources, but the efficiency of converting ecological benefits to economic benefits is still at a low level. On the one hand, this is due to the special management policies. At this stage, the value transformation of ecological products is still more dependent on the support of supply services and cultural services because the market transaction mechanism of material products and cultural products—such as travel and tourism—is more mature. However, the current management policy plays a significant role in limiting the value conversion of supply services and cultural services. Therefore, in nature reserves where development activities are strictly limited; realizing the regulation of ecosystems and pointing out the value of services is the focus of development. On the other hand, even without considering the management policy of the Qilihai area, ideally, the value conversion efficiency of 111.60% is still at a low level. The WWF’s *Earth Ecology Report 2004* states that the average global ecological footprint is 2.2 hectares, which means that it would take 2.2 Earths of production levels to meet current human needs. This can be used as a measure of the value conversion efficiency of ecological products. Among the designed value realization paths, the most effective is ecotourism, which should be a priority in development planning. Ecoagriculture and ecoforestry provide significant ecological resources for tourism development while providing considerable economic value. The ecological trading model and ecological compensation model, which are important means to realize the ecological value of the core area of the Seven Mile Sea wetlands, should be used as the main contribution to the transformation of the ecological value of the Seven Mile Sea wetlands into economic value in the future. However, the corresponding market mechanism and policy guarantee are not yet perfect. This is the work that needs to be further completed.

This study can provide a reference for decision-makers to assess the ecological value of the Qilihai area and to help identify significant ecological elements so that important ecological resources can be effectively protected and developed in the future development and utilization planning process in order to ensure the sustainable development of ecological resources in the Qilihai area. At the same time, it also provides a theoretical exploration of the future sustainable development of the Qilihai area. The simulation accounting based on the value realization paths can determine the development model with the highest value realization rate. It is recommended to make full use of the ecological resources of the Qilihai Wetland to drive the development of supply services and cultural services in the surrounding areas while focusing on exploring means of value realization such as compensation and trading to promote the value realization of regulation and support services.

## Figures and Tables

**Figure 1 ijerph-19-14575-f001:**
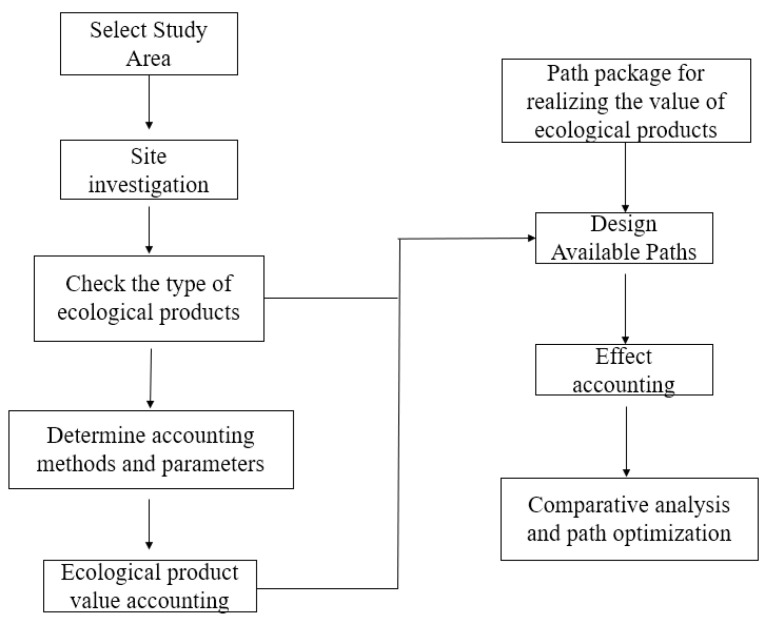
Technical Process.

**Figure 2 ijerph-19-14575-f002:**
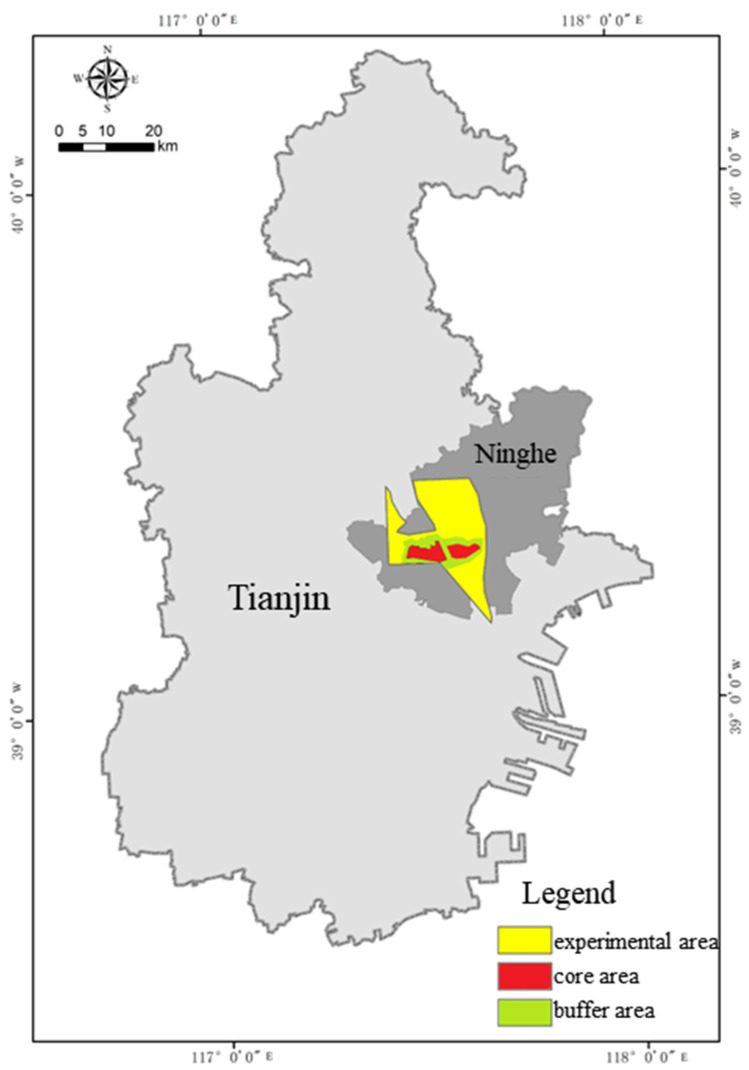
Geographic location map of Qilihai Wetland.

**Figure 3 ijerph-19-14575-f003:**
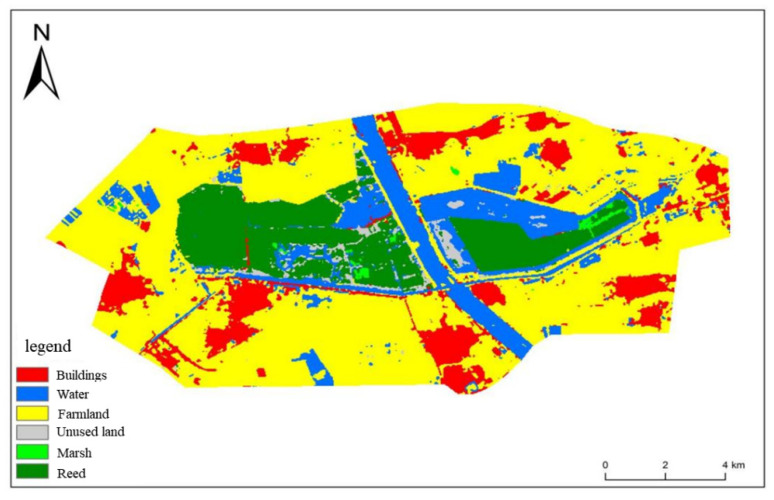
Classification of land-use types.

**Figure 4 ijerph-19-14575-f004:**
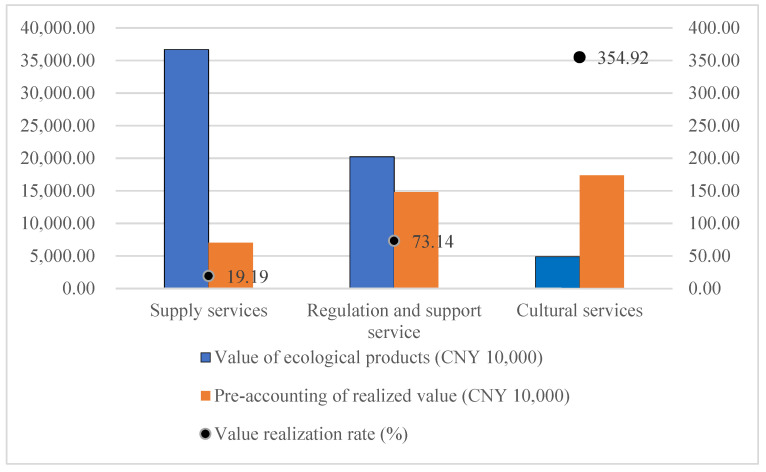
Ecoproduct value realization rate.

**Figure 5 ijerph-19-14575-f005:**
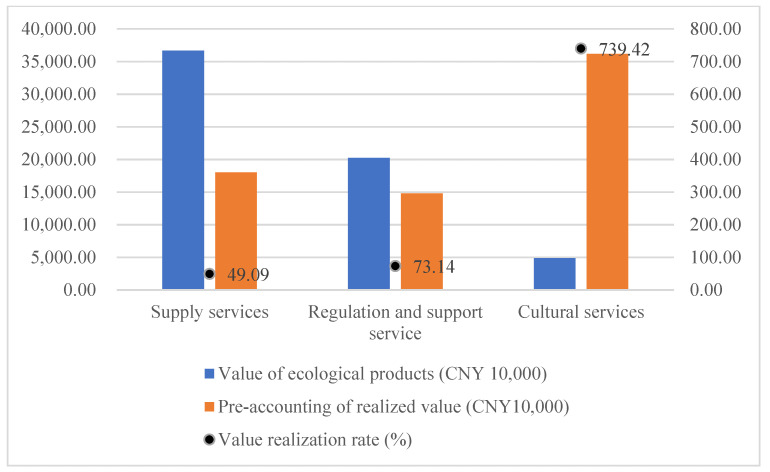
Ecological product value realization rate (considering the management policy of the Qilihai Wetland).

**Table 1 ijerph-19-14575-t001:** Typical cases of realizing the value of ecological products.

Value Realization Mode	Country	Value Realization Object	Case Content	Means of Realization
Ecological protection compensation	Multiple countries	National park	More than 100 international environmental protection organizations provide financial support to different national parks in the world—for example, Nature Conservancy Canada, the Canadian Nature Foundation, and the Sierra Club.	Financial compensation
Ecological industry development	Yanqing, China	/	Relying on climate conditions and resource endowments to optimize the industrial layout and promote the birth of new industrial models.	Ecotourism
Ecological equity transactions	USA	Pollution emission rights	The water pollution emission trading system economically stimulates nonpoint-source polluters to implement pollution control actions.	Emission trading
Transfer of resource property rights	Nanping, China	Forest ownership	Integrate and optimize fragmented management rights and use rights, form ecological assets, and transform ecological resource advantages into economic and industrial advantages.	Transfer and integration of management rights and use the right of forest rights
Ecological carrier premiums	Zhangzhou, China	Land price	Through planning and legislation, protect ecological resources in the form of crucial ecological boundaries, attract enterprises to improve the environment, and release ecological dividends through land premiums.	Land premiums
Resource quota transactions	USA	Wetland	Put forward the goal of “zero net loss” of wetlands, establish a market trading mechanism for wetland credit, and store and trade wetlands in the form of credit.	Wetlands are traded in the market in the form of credit.

**Table 2 ijerph-19-14575-t002:** Typical wetland development and utilization modes.

Wetland Development and Utilization Mode	Applicable Wetland Type	Classic CASE
Ecological sightseeing	Wetlands with high ecological value, high landscape value, and relatively far away from cities.	Shuntian Bay, South Korea
Popular science education	Wetlands with high science popularization value, artificial functional wetlands, or buffer zones of strictly protective wetlands.	Hong Kong Wetland Park
Urban leisure	Near the huge consumer market (mostly in or near the city), the wetland sensitivity is relatively low or, at least, the leisure activity area is not the core wetland reserve.	London Wetland Park
Compound development	Wetlands that are large in scale, have the conditions to develop into tourist destinations (with unique attractions, a good location, or strong capital), and have certain land resources.	Hangzhou Xixi National Wetland Park

**Table 3 ijerph-19-14575-t003:** Ecoproduct value assessment methods.

Value Category	Value Performance	Implication	Value Assessment Methodology
Supply Services	Material production value	Accounting for the market value of plant products and aquatic products.	Market Value Method V = ∑Si∙Yi∙Pi Si is the harvestable area of substance class i; Yi is the unit yield of substance class i; Pi is the market price of substance class i.
Regulation Services	Climate regulation	Absorbs CO_2_ and releases O_2._	Carbon sequestration value (silvicultural cost method) V = 1.63·R·Q·P_1_ + 1.19·Q·P_2_ R is the amount of carbon in CO_2_, i.e., 12/44; Q is the biomass of aquatic plants; P_1_ is the price of carbon sequestration; P_2_ is the price of oxygen.
	Flood storage function	Storage flood water.	Alternative Cost Method V = A·H·K A is the area of the wetland capable of storing water; H is the variation of wetland water level; K represents the cost of the reservoir per unit volume of water storage.
	Water purification	The growth of reeds can purify wastewater by first-hand use of nitrogen and phosphorus in the water.	Shadow Engineering Method V = Q·Y_N_·P_N_+ Q·Y_P_·P_P_ Qi is the annual yield of reed; Y_N_ is the N taken away per unit mass of reed harvested, P_N_ is the N removed from the investment per unit mass, Y_P_ is the P taken away per unit mass of reed harvested, and P^P^ is the P divided by the investment per unit mass.
Cultural Services	Tourism	A place for leisure, tourism, entertainment, etc.	Travel Expense Act V = Q·P Q is the number of tourist arrivals and P is the revenue benefit per tourist.
	Value of scientific research and education	Sites for scientific research and youth education.	Shadow Engineering Method V = P·A P is the wetland unit area research and education value; A represents the wetland area.
Support Services	Maintaining biodiversity	Provide a good habitat for various organisms to survive.	Results Reference Method V = P·A P is the ecological service value of biodiversity per unit area of wetland; A is the total local wetland area.
	Soil conservation function		Shadow price method, opportunity cost method V_11_ = ∑Ac·Ni·Pi, V_12_ = Ac·B/(1000·d·ρ) V_11_ is the unit value of soil nutrient retention; Ac is the soil retention; Ni is the pure content of soil nitrogen, phosphorus and potassium in the wetland; Pi is the price of fertilizer (urea, nitrogen-phosphorus-potassium compound fertilizer, diammonium hydrogen phosphate and potassium chloride); V_12_ is the economic benefit of reducing wasteland; Ac is the soil retention; B is the average annual return from agriculture; ρ is the soil capacity; and d is the soil thickness.

**Table 4 ijerph-19-14575-t004:** Value of ecological products in the Qilihai area.

Value Category	Secondary Classification	Value Volume (CNY 10,000)	Value Volume (USD 10,000)	Proportion
Supply Services	Aquatic Products	5920.00	815.14	10.40
	Plant Products	30,757.42	4235.04	54.05
	Subtotal	36,677.42	5050.18	64.45
Regulation Services	Atmospheric conditioning	626.78	86.30	1.10
	Flood water storage	2627.27	361.75	4.62
	Water Quality Purification	14,854.94	2045.40	26.10
	Subtotal	18,108.98	2493.46	31.82
Support Services	The generation and maintenance of biodiversity	1437.88	197.98	2.53
	Soil conservation	452.38	62.29	0.79
	Reduction in waste soil	229.82	31.64	0.40
	Subtotal	2120.08	291.92	3.73
Total		56,906.48	7535.55	100.00

**Table 5 ijerph-19-14575-t005:** Value of ecological products realized.

Paths		Value (CNY10,000)	Value (USD 10,000)	Proportion
Ecoindustry	Ecological agriculture	8020.18	1104.31	11.63
	Ecopremium	6505.14	895.70	9.43
	Ecoforestry	9983.38	1374.63	14.48
	Ecotourism	36,165.76	4979.73	52.44
	Subtotal	60,674.46	8354.37	87.98
Ecological compensation	Ecological compensation	2433.56	335.08	3.53
Ecological transaction	Resource Quota Trading	5826.30	802.23	8.45
	Ecological interest trading	30.91	4.26	0.04
	Subtotal	5857.21	806.49	8.49
	Total	68,965.22	9495.94	100.00

Note: The data of area and annual income in the table are kept in two decimal places, and the annual income per unit area is calculated according to the actual data.

## Data Availability

The data used to support the findings of this study will be available from the corresponding authors upon request.

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
