# Peer review of "Transformation Path of Ecological Product Value and Efficiency Evaluation: The Case of the Qilihai Wetland in Tianjin"

_ijerph, 2022, doi:10.3390/ijerph192114575_

Round 1

Reviewer 1 Report

The article deals with very important ecological issues combined with the assessment of their value. The authors point to the threats to the value of China's wetlands. They emphasize the value of wetland ecosystems. Interesting work, but a bit chaotic. Some threads appear in different parts of the text. The article needs to be read and corrected in this regard. The text is not well understood in places. I propose to subject the text to proofreading. In the abstract and in the introduction, the purpose of writing the article should be specified. What the authors want to achieve by writing this article.

Please show the use of the term "green product" not only in China, but also in the world. Enter who entered the date. Please state what is the new scientific value of the article. For a better understanding of the article, it is recommended to enter the value not only in Chinese currency, but converted into dollars or euros.

Good luck with your further work on the article

Reviewer

Reviewer 2 Report

Particular comments.

No figures are referenced in the text.

Tables 3, 4 and 5 are not cited in the text. It is difficult to follow the sequence of the manuscript if a table suddenly appears. It is not correct to leave to the reader the assumption of the usefulness of each table and figure presented.

Line 40-45. References are missing to validate the percentages mentioned.

The objectives should be clearly noted at the end of the introduction. It is difficult to identify the objectives of the study.

Figure 1, It is necessary to add latitude and longitude references.

Line 244. Seven specific paths are mentioned, but it is not clear in the text what they are. It is not clear that there are seven. Perhaps they are mentioned, but incorrectly numbered. Clarify this point.

Line 381. It is necessary to number the equation and give the reference, since there are many variants of the USLE formula.

Table 3 has no references. Is the way in which the Value Assessment Methodology is calculated a proposal of the authors? Or is it associated with any reference?

Line 451. The equation is not numbered and the units or dimensions of the variables in the equation are not defined. The same in lines 468, 489, 490, 491, 513.

Figure 3, presents the same values as table 4. Figure 3 should be deleted. The same is also the case with table 5 and figure 4. Figure 4 should be deleted.

Line 520. It seems to me that the authors have all the elements to divide and write individual sections for results and discussion. Actually the discussion starts on line 641.

General comments.

The material presented in the manuscript is of interest. The subject of the manuscript is of great importance globally and should be reproduced elsewhere. However, the methodological section is limited. The authors present an application of basic techniques already used before. It is difficult to find the scientific contribution of the manuscript; therefore it is difficult to accept that it is a scientific article. It is a technical note that tries to show the application of a methodology, in a very important region.

I suggest the authors to clearly show the objectives and the scientific contribution of their work. For example, in the title the word "prediction" is used and the evidence is not shown or is not clear in the development of what is presented.

I suggest the authors respond to the specific comments and reorganize the manuscript giving emphasis to the methodology part and present a very precise technical note with details that allow it to be reproduced anywhere.

Reviewer 3 Report

·      In the summary it says: This study helps assess the ecological value and important ecological elements of the Qilihai Wetland to ensure effective protection and development of important ecological resources and to achieve sustainable development of wetland resources and in results say This study was conducted to assess the value of ecological products in the Qilihai Wetland area and to account for the realization of the value of ecological products in a specific planning scenario.

·      In the section Research progress, It is hardly understandable what it refers to. If it refers to other studies, it would be better to integrate it into the introduction even though this would make it a very broad introduction, in itself it already is. Do you think this information is necessary? .

·      In the section Design of the path to realize the value of ecological products, should be clearer la descripción of Selection of study area

·      In the section Design of the path to realize the value of ecological products, should be clearer in the description

·      In section 2.4, Some sub-themes seem to have no relation to what is intended to be explained.

·      In the section Maintain soil nutrients and reduce waste soil: describes what might be part of a methodology, It could improve the order of the information and better describe this section (methodology).

·      The descriptionof the results should improve, it seems that a review of results from other authors is being done.

·      Discussion is not done with other authors and describes results orso it seems

·      Check if it is correct to put Exhibit 1 in the table.

·      In the conclusions it says: This study was conducted to assess the value of ecological products in the Qilihai Wetland area and to account for the realization of the value of ecological products in a specific planning scenario and the objective of the work is described in another way.

·      L 114 The results show that wetland ecosystems have important ecological service functions and great economic value, What results, Those of this research?

·      Conclusions, improve, as it is similar to results and discussion. They must be precise and mainly concluded based on the objectives of the manuscript

·      Manuscript is more like a review, it does not note that it is an investigation.

Round 2

Reviewer 2 Report

Second Revision

I have reviewed the authors' responses in detail. 

The objective of the paper was correctly written and included in the text.

The formulas were correctly numbered.

The authors removed the requested figures. The necessary data were added in the corresponding tables.

The methodology was detailed as requested. It was complemented with the diagram shown in Figure 1. The authors did an excellent job in clarifying the procedure. Good work.

The discussion was successfully included.

Units were added to the variables in all equations.

The above are the main modifications made by the authors.

In my opinion, the authors reviewed and addressed all comments scrupulously.

The manuscript is now suitable for publication.

Author Response

Thank you very much for your advices on our work. We are glad that the revised version can get your approval.

Reviewer 3 Report

Los autores escucharon los comentarios y estoy satisfecho.

Author Response

(The authors gave the same response as above.)
